# The Effects of Social Determinants and Resilience on the Mental Health of Chilean Adolescents

**DOI:** 10.3390/children10071213

**Published:** 2023-07-13

**Authors:** Alejandra Caqueo-Urízar, Patricio Mena-Chamorro, Diego Henríquez, Alfonso Urzúa, Matías Irarrázaval

**Affiliations:** 1Instituto de Alta Investigación, Universidad de Tarapacá, Arica 1000000, Chile; 2Centro de Justicia Educacional CJE, Pontificia Universidad Católica de Chile, Santiago 7820436, Chile; pmena@uta.cl (P.M.-C.); xdiegohenriquez@gmail.com (D.H.); 3Escuela de Psicología, Universidad Católica del Norte, Antofagasta 1270709, Chile; alurzua@ucn.cl; 4Department of Psychiatry, Faculty of Medicine, Clinical Hospital, Universidad de Chile, Santiago 8380453, Chile; mirarrazavald@uchile.cl; 5Millennium Institute for Research in Depression and Personality, MIDAP, Santiago 8380453, Chile

**Keywords:** adolescents, social determinants, resilience, vulnerability, mental health

## Abstract

The aim of this research was to evaluate the effects of social determinants (i.e., gender, educational vulnerability, and socioeconomic status) and resilience on the mental health of Chilean adolescents in pre-, during, and post-COVID-19 pandemic contexts. The study included a group of 684 students, ranging in age from 12 to 18 years, who were attending educational institutions in the city of Arica. The Child and Adolescent Assessment System (SENA) was used to measure mental health problems, the Brief Resilience Scale for Children and Youth (CYRM-12) was used to measure resilience, and the Vulnerability Index of Educational Institutions was used to measure educational vulnerability. The results suggest increases in depressive, anxious, and social anxiety symptomatologies over time (wave by year, 2018, 2020, and 2021). In addition, multiple linear regression models showed predictive effects of the COVID-19 pandemic, gender, vulnerability index, socioeconomic status, and resilient behaviors on mental health problems. The worsening of mental health indicators over time requires the greater coordination and integration of mental health experts in the most vulnerable educational centers.

## 1. Introduction

In recent years, the mental health of millions of people has systematically worsened [1]. In 2019, 970 million people reported living with a mental disorder, with anxiety disorders and depression being more prevalent [2]. In 2020, the number of people living with a mental disorder increased significantly due to the COVID-19 pandemic [1]. Although effective prevention and treatment options exist, access to effective care is limited for most people with mental disorders [3].

The social context in which human beings develop plays a fundamental role in the well-being and care for people’s mental health [4]. The high frequency of occurrence and severity of mental disorders are strongly correlated with individuals’ socio-economic circumstances, including factors such as poverty, income inequality, or involuntary migration [5,6]. The social determinants of mental health that stand out the most in the literature are demographic, which include gender, age, ethnicity, and life expectancy [7,8,9,10,11]; economic, where financial and employment status, housing, and income inequality are considered [12,13,14,15]; an individual’s area of residence, which is related to variables such as safety, the availability of services, and places of recreation [16,17,18,19]; environmental events, such as natural disasters and armed conflicts, as well as hazards to the ecosystem due to climate change [20,21,22]; and the sociocultural domain, which encompasses education, interpersonal relationships, social capital, culture, and a person’s sense of belonging in their community [23,24,25].

Given the influence of social determinants on mental health problems, it has been shown that for a group of subjects, exposure to the same traumatic and/or adverse event will produce different reactions and/or responses [26]. One of these responses is the resilient response which, in other words, implies a positive adaptation to adversity [27,28] that reduces anxious and depressive symptoms [29,30,31,32]. People’s resilient responses are influenced by individual and family factors [33,34]. Also, the presence of resilient behaviors has been demonstrated in adolescents who have been exposed to various adversities and traumas [35]. Despite the relevance of resilience as a protective factor for mental health problems [36], few studies have focused on studying the effects of resilience in conjunction with social determinants of mental health in Latin American adolescents [37,38]. The literature suggests that most mental health problems begin in the first years of life [39,40], during adolescence, which is a phase characterized by significant vulnerability, when negative social encounters have the potential to detrimentally impact the cognitive, emotional, and behavioral growth of individuals [39]. Likewise, the infant–juvenile population is also affected by social determinants. Female adolescents have higher risks of developing depressive and anxiety disorders [41,42], while males are associated with a greater likelihood of developing a substance use disorder [8,43,44]. Likewise, children and adolescents from low socioeconomic backgrounds encounter significant disparities in education, social opportunities, and healthcare [45], making them two to three times more susceptible to developing mental health issues compared to their counterparts from higher socioeconomic strata [46]. In addition, adolescents who are exposed to high levels of community violence, the deprivation of services and a scarcity of recreational areas in residential neighborhoods in this population have shown higher prevalences of psychotic and depressive disorders, as well as greater likelihoods of substance abuse and bullying [16,17,47]. In addition, it has been observed that family relationships also have an important impact on the development of mental disorders in children and adolescents, as in the case of anxious and depressive disorders [48,49], while friendship relationships and social support can decrease these possibilities, establishing themselves as protective factors [50].

On the other hand, COVID-19 has had a significant impact on mental health worldwide [31]. The pandemic and measures of confinement and social distancing have created a stressful and challenging environment that has exacerbated existing mental health problems and resulted in new disorders in many individuals [27]. Depression and anxiety are two of the most common disorders affected by the pandemic due to uncertainty about health, fear of infection or contagion, loss of employment, and social isolation [36,37]. In addition, social anxiety has also been affected due to social distancing and the use of masks, which has hindered social interactions and increased feelings of loneliness and isolation [30,37]. Despite these challenges, the resilience of many people has been remarkable as they find ways to cope and maintain their mental health [26,27,28].

In Chile, the available findings point in the same direction as those presented in other international contexts [44,51,52]. For example, studies show that the most recurrent health problems in the infant–juvenile population in northern Chile are anxiety, depression, and conduct disorders, and the female population turns out to be the most affected compared to males [44,51]. Economic inequality has proven to be a very relevant factor influencing the mental health of adolescents since the educational level, access to health care, and psychosocial problems faced by individuals this population will depend on the socioeconomic status of their family. Chile’s economic and social approach promotes the privatization of essential services, which hinders equal access to these services. Government policies during the COVID-19 pandemic maintained extensive and demanding confinement measures in both labor and school environments. This situation had a negative impact on the upbringing of the most disadvantaged children, as well as on their parents’ abilities to meet their children’s emotional, educational, and health needs. As a result, these children became more vulnerable when facing challenges relating to their mental health [44,53,54,55].

Despite the existence of some studies contrasting the effects of the social determinants of adolescent mental health in the Chilean context, the research is still insufficient and/or scarce. Moreover, no study has jointly contrasted the effects of the COVID-19 pandemic, resilience, and social determinants on the mental health of young people. Hence, the objective of this research was to assess the impact of social determinants and resilience on the mental well-being of Chilean adolescents within the context of the COVID-19 pandemic, before, during, and after its occurrence.

## 2. Materials and Methods

### 2.1. Participants

A repeated cross-sectional design [56] was used with three samples of high school students belonging to urban educational establishments in 2018, 2021, and 2022. Data collection was performed via a non-probability convenience sampling strategy [57]. The 2018 sample was composed of 249 students, of which 52.2% (*n* = 130) attended educational establishments with low vulnerability indexes and 47.8% (*n* = 119) attended educational establishments with high vulnerability indexes. The ages of the students ranged from 12 to 18 years, with a mean age of 14.42 (SD = 1.83); 50.2% (*n* = 125) reported being female, 90% (*n* = 224) reported being Chilean, 70.3% (*n* = 175) reported belonging to a religion, and 67.1% (*n* = 167) reported no ethnicity (the term no ethnicity refers to the absence of a specific ethnic or racial identity. This option is for individuals who do not identify with any particular ethnic group or prefer not to disclose their ethnicity). The 2021 sample was composed of 206 students, with 47.6% (*n* = 98) attending low-vulnerability-index educational institutions and 52.4% (*n* = 108) attending high-vulnerability-index educational institutions. The ages of the students ranged from 12 to 18 years, with a mean age of 14.23 (SD = 1.59); 55.3% (*n* = 114) reported being female, 87.4% (*n* = 180) reported being Chilean, 55.3% (*n* = 114) reported belonging to a religion, and 58.7% (*n* = 121) reported having no ethnicity. The 2022 sample was composed of 229 students in which 40.2% (*n* = 94) attend low-vulnerability-index educational institutions, and 59.8% (*n* = 140) attend high-vulnerability-index educational institutions. The ages of the students ranged from 12 to 18 years, with a mean age of 14.48 (SD = 1.55), 56.3% (*n* = 129) reported being female, 88.0% (*n* = 206) reported being Chilean, 56.5% (*n* = 130) reported belonging to a religion, and 57.1% (*n* = 129) reported having no ethnicity. The sociodemographic details are presented in Table 1.

### 2.2. Instruments

Ad-hoc sociodemographic questionnaire: This questionnaire collected information on the students’ sex, age, socioeconomic status, nationality, religiosity, and ethnicity. The year of data collection was used as a proxy for the effect of the COVID-19 pandemic.

Vulnerability index of educational institutions: This index was calculated by estimating the weighted percentage of the risk needs of students attending different educational institutions; some of the factors encompassed in this category are limited maternal education, unaddressed medical necessities, existential needs, insufficient weight for age, and other related aspects. According to the statistics presented in the Annual Municipal Development Plan of Arica [58], a classification of “high” vulnerability and “low” vulnerability was made. According to this report, between 2014 and 2017, public schools exhibited an average vulnerability index of 86%, surpassing the 77% vulnerability level observed in the commune [59]. Conversely, during the same timeframe, subsidized and private schools had an average vulnerability index of 74%, which remained lower than the percentage observed at the communal level [59]. The cut-off point was the average vulnerability percentage of the commune. The most elevated score in the low vulnerability category was 73, while the lowest score in the high vulnerability category was 78.

Child and adolescent assessment system (Sistema de Evaluación de Niños y Adolescentes, SENA): This instrument measures several emotional and behavioral problems [60]. Self-report versions were used for high school students 12 to 18 years old. The high school version was composed of 188 items; however, for this study, only the items pertaining to the internalized problem dimensions (e.g., depression, anxiety, and social anxiety, 32 items in total) were used. The items were on a five-point Likert behavioral/attitudinal statement scale (from 1 = “never or almost never” to 5 = “always or almost always”). Higher scores suggest the presence of a higher level of maladaptation and difficulties in adapting to the context. The version used in this study reported evidence of reliability and validity based on the internal structure of the test (Sánchez-Sánchez et al., 2016). The SENA scales have presented Cronbach’s alpha coefficients greater than 0.75 [61].

Brief Resilience Scale for Children and Youth (CYRM-12) [62]: This scale measures the degree of resilience in the face of adversity based on the interaction between individual, relational, community, and cultural factors (e.g., “I try to finish what I start” and “My family will be there for me in difficult times”). Response options respond to behavioral statements on a 5-point Likert scale (from 1 = “never” to 5 = “very much”). Higher scores suggest higher levels of resilience. Llistosella et al. [63] translated and validated the 32-item version of the Child and Youth Resilience Measure (CYRM) into Spanish and reported evidence of validity of their scores, reporting a satisfactory internal structure of the test and satisfactory reliability coefficients (α > 0.80).

### 2.3. Procedures

The study, which is part of a larger project from the Educational Justice Center, was approved by the Ethics Committee of the Universidad de Tarapacá (26–2017) on 20 September 2017. The researchers reached out and extended voluntary invitations to principals and counselors from educational institutions in Arica to partake in this study. Out of the 42 schools invited, 29 principals agreed to participate in the data collection for the years 2018, 2021, and 2022, resulting in a refusal rate of 31%.

Parental consent was sought after explaining the purpose and scope of the study. The students then signed a form agreeing to participate. The questionnaires were administered in a pencil and paper format by at least 2 trained assistants. The application procedure took place within each class for every course, in which a minimum of two trained interviewers, along with the respective teachers of the courses, responded to the questionnaire. The duration was approximately 45 min, and data collection sessions were conducted from March to December of 2018, 2021, and 2022.

### 2.4. Data Analysis

Initially, to characterize the sample in each year, the proportion of the students’ sociodemographic variables was obtained and the central tendency (i.e., mean), dispersion (i.e., standard deviation, minimum, and maximum), and shape (i.e., skewness and kurtosis) of the continuous variables were obtained. The Shapiro–Wilk test was also used to assess univariate normality.

Afterward, a statistical technique called an analysis of variance (ANOVA) was utilized to assess and compare the variations in the average scores of the depression, anxiety, and social anxiety scales across different years of application (serving as a proxy to gauge the impact of the COVID-19 pandemic). Multiple comparisons were conducted with a Games–Howell correction because the data did not show homoscedasticity (depression: Levene’s test, F = 4.64, *p* = 0.010; anxiety: Levene’s test, F = 5.24, *p* = 0.006; and social anxiety: Levene’s test, F = 9.52, *p* < 0.001), and the partial eta squared (η2p) was used as an estimate of effect size. Parametric analyses were used because ANOVAs are sufficiently robust with data that did no present homoscedasticity and are not normally distributed either (see Table 2) [64].

Finally, to evaluate the potential predictive capacities of social determinants and resilience regarding students’ mental health indexes, multiple linear regression analyses were performed. One model was estimated for depression, another for anxiety, and a final model for social anxiety. Gender, the year of application (i.e., used as a proxy to measure the effect of the COVID-19 pandemic), vulnerability index, socioeconomic status (SES), nationality, ethnicity, being Aymara (The Aymara are an indigenous ethnic group with a strong presence in the Andean region that includes the territories of Chile, Peru, and Bolivia), religion, and resilience were included as predictor variables. The categorical predictor variables were transformed into dummy variables. The standardized ß coefficients represented changes in the standard deviations of the criterion variables. Predictor variables with higher standardized ß coefficients suggest a greater relative effect on student mental health indexes. All assumptions were met. The presence of multicollinearity among predictor variables was ruled out via the inflated variance factor (IVF), for which the values were less than 5 for all variables. The residuals were independent of each other (depression: Durbin–Watson statistic, DW = 2.02, *p* = 0.880, anxiety: DW = 2.03, *p* = 0.782, social anxiety: DW = 2.03, *p* = 0.722). The presence of homoscedasticity was verified by examining a scatter plot of the predictors and standardized residuals. The normality of the residuals for each dependent variable was assessed via the examination of a histogram and a Q-Q plot of the standardized residuals. The statistical hypothesis testing for the data analyses was conducted at a significance level of 5%. The statistical analyses were carried out using Rstudio and Jamovi version 2.3.2 software.

## 3. Results

### 3.1. Descriptive Analysis

Symmetry and kurtosis were outside the acceptable ranges to be considered normally distributed in the 2018 and 2021 collections, with the exception of 2022. In addition, the Shapiro–Wilk test showed that none of the mental health variables had a normal distribution in the 2018, 2021, and 2022 collections. Details of the descriptive analyses of the study variables are presented in Table 2.

The ANOVA showed statistically significant differences between the different years of application and the mental health indexes. Regarding the dependent variable of depression, the main effect of application year was statistically significant (F (2, 443) = 19.24, *p* < 0.001). Post-hoc test results showed a significant increase in depression scores between the application years 2018 and 2021 (M = 2.11, SD = 0.86, M = 2.47, SD = 1.01, t = −4.02, *p* < 0.001, respectively), as well as 2018 and 2022 (M = 2.59, SD = 0.92, t = −5.97, *p* < 0.001). There were no statistically significant differences between 2021 and 2022 in depression scores (t = −1.36, *p* = 0.362). According to the dependent variable of anxiety, the main effect of tge year of application was also statistically significant (F (2, 441) = 25.21, *p* < 0.001). Post-hoc test results showed a significant increase in anxiety scores between the application years 2018 and 2021 (M = 2.48, SD = 0.84, M = 2.92, SD = 1.00, t = −4.94, *p* < 0.001, respectively), as well as 2018 and 2022 (M = 3.04, SD = 0.96, t = −6.64, *p* < 0.001). There were no statistically significant differences between 2021 and 2022 in depression scores (t = −1.23, *p* = 0.434). Finally, regarding the dependent variable of social anxiety, the main effect of the year of application was also statistically significant (F (2, 438) = 20.91, *p* < 0.001). Post-hoc test results showed a significant increase in social anxiety scores between the application years 2018 and 2021 (M = 2.38, SD = 0.77, M = 2.79, SD = 0.98, t = −4.86, *p* < 0.001, respectively), as well as 2018 and 2022 (M = 2.82, SD = 0.89, t = −5.78, *p* < 0.001). There were no statistically significant differences between 2021 and 2022 in depression scores (t = −0.34, *p* = 0.935).

### 3.2. Multiple Linear Regression Models on Mental Health Indexes

The results of the multiple linear regression analyses showed that for the depression criterion variable, the model was statistically significant (F (11) = 23.8, *p* < 0.001) and able to explain 27.3% of the variance in the students’ depression scores. There is a significant and negative relationship between depression and resilience scores when all other variables remain constant (i.e., their value was zero). Thus, with an increase in one standard deviation of resilience, the depression score will decrease on average by −0.706. Depression scores increased on average by 0.247 and 0.592 when the application years were 2021 and 2022, compared to 2018 when, all other variables remained constant. The depression scores decreased on average by −0.424 when students were male compared to female when all other variables remained constant. The depression scores increased on average by 0.204 in students belonging to families with medium SESs compared to a low SES when all other variables were held constant. The details of the multiple linear regression are presented in Table 3.

For the anxiety criterion variable, the multiple linear regression model was statistically significant (F (11) = 15.54, *p* < 0.001) and able to explain 19.3% of the variance in the students’ anxiety scores. There is a significant and negative relationship between anxiety and resilience scores when all other variables are held constant. Thus, with an increase in one standard deviation of resilience, the anxiety score will decrease on average by −0.473. Anxiety scores increased on average by 0.385 and 0.780 when the application years were 2021 and 2022 compared to 2018 with all other variables held constant. Anxiety scores decreased on average by −0.548 when the students were male compared to female, with all other variables remaining constant. Anxiety scores decreased on average by −0.183 in students belonging to families with low EVIs compared to a high EVI when all other variables remained constant. The details of the multiple linear regression are presented in Table 3.

For the social anxiety criterion variable, the multiple linear regression model was statistically significant (F (11) = 12.76, *p* < 0.001) and able to explain 16.2% of the variance of students’ social anxiety scores. There is a significant and negative relationship between social anxiety and resilience scores when all other variables are held constant. Thus, with an increase in one standard deviation of resilience, the social anxiety score will decrease on average by −0.386. Social anxiety scores increased on average by 0.302 and 0.644 when the application years were 2021 and 2022, compared to 2018, with all other variables held constant. Social anxiety scores decreased on average by −0.534 when students were male compared to female, with all other variables remaining constant. Social anxiety scores increased on average by 0.181 in students belonging to families with medium SESs compared to a low SES when all other variables were held constant. The details of the multiple linear regression are presented in Table 3.

## 4. Discussion

The purpose of the present study was to evaluate the effects of social determinants and resilience on the mental health of Chilean adolescents in pre-, during-, and post-COVID-19 pandemic contexts. This study showed that students attending schools with high and low vulnerability indexes experienced significantly worsened mental health over the years from 2018 (i.e., pre-pandemic) to 2021 (i.e., during the pandemic) and 2022 (i.e., post-pandemic). This finding is consistent with the ample evidence of the negative effects of the COVID-19 pandemic on adolescent mental health [65,66,67,68,69,70,71,72,73]. Therefore, it is necessary to design interventions that provide tools to facilitate the return to the classroom, especially in context of vulnerability.

This study also showed increases in depressive, anxious, and social anxiety symptomatologies when the students were female, their family had a medium SES (i.e., only in depression and social anxiety), and they belonged to educational establishments with high vulnerability indexes (i.e., only in anxiety), while the resilient response of the students was able to reduce mental health problems. These findings are also consistent with those found in the literature [29,30,31,32,37,38], with the exception of the socioeconomic status factor, which may seem contradictory. One might expect worse mental health in families with low socioeconomic levels since it is usually assumed that families with low economic resources possess greater vulnerability [74]. However, it is important to note that in this study, the variable SES was constructed in terms of the schools attended by the students (i.e., public, subsidized, and private schools). Therefore, there is a possibility that the participation of families with low and medium SESs between public and subsidized schools is not very clear. It is possible that the majority of families with low SESs are participating in subsidized schools and vice versa. In this sense, the vulnerability index could complement this contradictory finding. Similar to how numerous public schools were categorized as having a low vulnerability index, certain subsidized schools were identified as having a high vulnerability index. As a result, these outcomes partially align with previous findings documented in the existing literature [12,13,14,15,23,24,25,43,53,54,55].

Some factors that may support these results probably respond to the difficult access that students belonging to families from low and middle socioeconomic strata have to treatment, the negative effects of the COVID-19 pandemic, and the perceptions of uncertainty, loneliness, and/or of not possessing the necessary tools to perform adequately in the neoliberal system that prevails in Chile [43,53,75,76]. The existence of disparities in anxiety levels, according to the vulnerability index in school communities, could indicate the importance of enriching standardized and generalist interventions. This would imply adapting these interventions to the specific context of vulnerability faced by adolescents.

This study has some limitations. First, a non-probabilistic sampling strategy was used, which limits the generalizability of the results to other sociocultural contexts. Secondly, only students from one region of northern Chile participated; therefore, future research should contrast these results with students from other regions of the country. Third, although a repeated cross-sectional design was used which allows for the establishment of pseudo-longitudinal explanations [77], it is not possible to establish changes over time in the students (i.e., trajectories); therefore, future studies should use longitudinal designs that allow the findings of this study to be contrasted with those in a sample of adolescents measured at different times. Finally, this study did not consider other sources of information that contribute to a better understanding of adolescent mental health, such as information reports from parents, guardians, and/or teachers.

Despite the limitations, the findings of this study provide relevant information about the resilient responses of high school students in Chile attending low- and high-vulnerability educational establishments pre-, during, and post-COVID-19 pandemic and the pandemic’s effects on their mental health. The results suggest the design of interventions with new perspectives based on the students’ context of vulnerability. It also provides background information for future research to include variables not considered in this study and to test more complex explanatory models. Likewise, the findings of this study can be used as inputs to propose new orientations in the design of public policies that produce structural changes in the reduction of the mental health needs of Chilean high school students. Finally, it is suggested to strengthen the coordination between mental health experts and the most vulnerable educational centers, implement programs to promote resilience, address socioeconomic and educational inequalities, promote a comprehensive health approach in the educational curriculum, and establish psychological and emotional support programs within schools. These suggestions aim to address social determinants and promote mental health in the educational context to improve student well-being.

## Figures and Tables

**Table 1 children-10-01213-t001:** Sociodemographic characteristics of the samples.

		2018 (*n* = 249)	2021 (*n* = 206)	2022 (*n* = 229)
		M (SD) ± Range or *n* (%)	M (SD) ± Range or *n* (%)	M (SD) ± Range or *n* (%)
Sex				
	Female	125 (50.2%)	114 (55.3%)	129 (56.3%)
	Male	124 (49.8%)	92 (44.7%)	100 (43.7%)
Age		14.42 (1.83) ± 12–18	14.23 (1.59) ± 12–18	14.48 (1.55) ± 12–19
EVI				
	Low vulnerability	130 (52.2%)	98 (47.6%)	94 (40.2%)
	High vulnerability	119 (47.8%)	108 (52.4%)	140 (59.8%)
SES				
	Low	92 (36.9%)	48 (23.3%)	144 (61.5%)
	Middle	127 (51.0%)	158 (76.7%)	71 (30.3%)
	High	30 (12.0%)	0 (0.0%)	19 (8.1%)
Nationality				
	Chilean	224 (90%)	180 (87.4%)	206 (88.0%)
	Foreigner	25 (10%)	26 (12.6%)	28 (12.0%)
Religion				
	With religion	175 (70.3%)	114 (55.3%)	130 (56.5%)
	Without religion	74 (29.7%)	90 (43.7)	100 (43.5%)
	Not reported	0 (0%)	2 (1%)	0 (0%)
Ethnic group				
	With ethnicity	82 (32.9%)	85 (41.3%)	97 (42.9%)
	No ethnicity	167 (67.1%)	121 (58.7%)	129 (57.1%)
Aymara				
	Aymara	58 (23.3%)	68 (33.0%)	75 (33.2%)
	No-Aymara	191 (76.7%)	138 (67.0%)	151 (66.8%)

M = mean; *n* = number of subjects; SD = standard deviation. EVI = Educational Vulnerability Index; SES = socioeconomic status.

**Table 2 children-10-01213-t002:** Descriptive statistics of the dependent variables of the study.

	M	SD	Min–Max	S ^a^	K ^a^	Shapiro–Wilk	*p*
2018							
Resilience	3.87	0.56	2.1–5.0	−2.90	−0.21	0.981	0.002
Depression	2.11	0.86	1.0–4.8	6.47	1.13	0.908	<0.001
Anxiety	2.48	0.84	1.1–5.0	5.18	0.37	0.944	<0.001
Social anxiety	2.38	0.77	1.0–4.6	4.14	0.42	0.962	<0.001
2021							
Resilience	3.73	0.66	1.8–5.0	−3.27	−0.12	0.971	<0.001
Depression	2.47	1.01	1.0–5.0	3.57	−1.53	0.946	<0.001
Anxiety	2.92	1.00	1.0–5.0	1.53	−2.14	0.973	<0.001
Social anxiety	2.79	0.98	1.0–5.0	2.11	−1.74	0.970	<0.001
2022							
Resilience	3.69	0.61	2.8–5.0	−1.42	−1.08	0.989	0.064
Depression	2.59	0.92	1.0–5.0	1.77	−2.28	0.973	<0.001
Anxiety	3.04	0.96	1.0–5.0	−0.13	−2.95	0.977	<0.001
Social anxiety	2.82	0.83	1.0–4.9	1.55	−2.74	0.969	<0.001

**^a^** Standardized coefficient.

**Table 3 children-10-01213-t003:** Results of the multiple regression analysis.

Variables	Depression		Anxiety		Social Anxiety	
F(df)	23.8 (11)		15.54 (11)		12.76 (11)	
*p*	<0.001		<0.001		<0.001	
Adj. R^2^	0.273		0.193		0.162	
	Standardized coefficients and value *p*					
	β	*p*	β	*p*	β	*p*
Intercept	4.699	---	4.307	---	3.92	---
Resilience	−0.706	**<0.001**	−0.473	**<0.001**	−0.386	**<0.001**
2021	0.247	**0.002**	0.385	**<0.001**	0.302	**<0.001**
2022	0.592	**<0.001**	0.780	**<0.001**	0.644	**<0.001**
Men	−0.424	**<0.001**	−0.0548	**<0.001**	−0.534	**<0.001**
High EVI	0.024	0.762	−0.183	**0.033**	0.018	0.826
Middle SES	0.204	**0.016**	0.147	0.104	0.181	**0.036**
High SES	0.204	0.164	0.161	0.309	0.099	0.509
Migrant	−0.077	0.449	−0.131	0.230	0.065	0.529
No ethnicity	−0.062	0.570	0.045	0.705	−0.115	0.309
No Aymara	0.101	0.392	0.015	0.904	−0.113	0.352
Without religion	−0.054	0.422	−0.081	0.265	−0.013	0.846

Note: F = F statistic; *p* = Significance; adj. R^2^ = corrected R-squared coefficient; β = standardized regression coefficient; EVI = Educational Vulnerability Index; SES = Socioeconomic Status. Bold values indicate statistical significance (*p* < 0.05).

## Data Availability

The data presented in this study are available upon request from the corresponding author.

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
