# Peer review of "The Effects of Social Determinants and Resilience on the Mental Health of Chilean Adolescents"

_children, 2023, doi:10.3390/children10071213_

Round 1
Reviewer 1 Report
I read the article by Alejandra Caqueo-Urízar et al. entitled «Social determinants and resilience in the mental health of adolescents in educational establishments of low and high socioeconomic vulnerability in Chile". This is an interesting topic and a well-written article.
Here are my comments
-It would be particularly interesting and necessary for the authors to provide data on the pandemic during the conduct of the study. Were government protection measures for COVID- 19 in place? Was there an intense pandemic wave?
-I think the sample size is large enough to be covered by the central limit theorem. The authors need not worry so much about the normal distribution of factors. It's enough for the authors to argue a little better for the randomness of sample selection. On the other hand the necessary conditions for conducting regression analyses, I think are met.
-It is unfortunate that the authors did not do a mediation analysis, but I certainly don't think it is necessary.
Author Response
In attachment. Thank you for your comments.

Reviewer 2 Report
As a study of the changed mental health situation of young people in one community that may be directly related to pre-, pandemic, and post-pandemic conditions they experienced, this paper has substantial potential. Your review of literature related to social determinants of mental health is appropriately focused, and your inclusion of age, vulnerability, gender, and personal social characteristics as major variables is appropriate. The choice of resilience as a major variable is also central to the design and execution of the study. There are some terms that are central to the study that need further elucidation, and a few sections that could benefit from editing. The presentation of results is generally clear and compelling. Also the discussion draws appropriate conclusions, but would benefit from further elaboration in a few crucial places. Some specific suggestions for revision follow for each section.
Abstract
Suggest that you specify what the social determinants are for this study, and how measured. Also in lines 19-20, do you mean that you’ve observed an increase in mental health symptoms over time? Please clarify what you mean by “The observed increase in mental health indices…” Also in lines 23-24, are you referring to high levels of socioeconomic vulnerability?
Keywords –Mental health appears twice.
Introduction
p. 1, line 35. Not sure what you mean by “In this sense”
p. 2, lines 62-66. Please revise to clarify. Do you mean that children from low-income families are two to three times more likely to develop mental health problems from those from families with higher incomes?
p. 2, lines 72-23. Reference 50 appears to be about adult women who have just given birth to children., not about children and youth.
p. 2, line 82-83. Should this be revised to specify “…as well as on their parents’ ability to meet their emotional, educational, and health needs.” ?
p. 2, line 88. Suggest you revise to clarify that no prior study has examined the relationship between the COVID 19 pandemic, resilience, social determinants, and youth mental health.
Materials and Methods
p. 3, paragraph 1. Please specify for readers not familiar with Chile, what is meant by the statement that youth indicated that they have no ethnicity?
p. 3-4 Table 1 At this point, there has been no definition given for NSE. Please indicate the meaning of this abbreviation, which also appears in Table 3..
p. 4, Table 1. Please indicate in the text describing participants what the Aymara subgroup is.
p. 5, lines 160-161. Suggest that you revise this to say “The duration was approximately 45 minutes and data collection sessions were conducted from….”
Results
p. 7, Table 3. It would be helpful to indicate that the vulnerability index of schools in involved in the designations (High IVE, middle NSE, High NSE). Also what is IVE?
Discussion
p. 8, lines 292-294, It would be helpful to give a more detailed statement that clarifies what is meant by “the need to adapt and complement universal interventions to the context of vulnerability in which adolescents are exposed.”
p. 8, lines 313-315. Please elaborate on this statement. Do you have suggestions about policies that might help improve mental health of Chilean high school students?
This manuscript presents compelling reasons for undertaking a study that examines the factors affecting the mental health of students in one region of Chile during times including pandemic experiences, clearly states the methods used in the study, provides clear and convincing results and makes important points about the implications of the results. The English language used is generally clear, direct, and appropriate.
Author Response
In attachment. Thank you for your review.

Reviewer 3 Report
Thank you very much for your submission to this Journal. Your paper is well-written and organized. However I have few minor concerns that may make it cleared to the readers.
1. Title:
a) I think it is better to change "in: to "on"; Social determinants and resilience on the mental health,,,,,
b) it is a bit confusing when using (Low and high socioeconomic vulnerability); because in the text you have two different things in regard to socioeconomic status and vulnerability index. In this form it appears that they are one variable.
2. Introduction
a) in line 31, add "with mental disorders" after people.
b) I do not see any mentioning for the effect of covid-19 on your major variables. It is worthy to mention that covid-19 affected mental health issues especially depression, anxiety, and social anxiety as well as resilience.
c) It is also better to add a paragraph about the educational system in Chilean schools in terms of types (subsidized, private, and public, ...etc..) and reflect that when discussing your results. The readers from out side Chile is not familiar with types of schools and their relation to vulnerability index.
3. Materials and methods
a) participants: lines 100, 105, and 110; do not start sentences with numbers (50.2% and 55.3% and 56.3%).
b) Table 1. What is NSE (confused with socioeconomic status, are they the same thing)? There are also few abbreviations that not identified in the first appearance.
c) What is Aymara? Did not mentioned in the introduction and did not mentioned in the demographics or social determinants. This concept maybe familiar to Chilean but not all readers. You only mentioned that in line 182 and in the table.
d) Instruments: 1) in line 125, clarify whether 77% of the commune is for general people in the world or for Chilean citizens. 2) indicate how many items were used in the the scales: SENA and CYRM.
e) In line 189 indicate the meaning of DW.
4. Results: organized, complete, and adequate. Written very well. Just two notes: in line 245 you have EVI, and in the table you have IVE. Also, for table 3 indicate the meaning of IVF and NSE.
5. Discussion: Just convince the readers that the changes in your result due to covid-19.
6. References: Some of the non-classical references are too old. 1996, 2005, 2006, 2011, 2010. Reconsider including them or not.
Author Response

(The authors gave the same response as above.)
